# Learning A Structured Optimal Bipartite Graph for Co-Clustering

**Feiping Nie**[1], **Xiaoqian Wang**[2], **Cheng Deng**[3], **Heng Huang**[2]*

[1] School of Computer Science, Center for OPTIMAL, Northwestern Polytechnical University, China
[2] Department of Electrical and Computer Engineering, University of Pittsburgh, USA
[3] School of Electronic Engineering, Xidian University, China
feipingnie@gmail.com,xqwang1991@gmail.com
chdeng@mail.xidian.edu.cn,heng.huang@pitt.edu

## Abstract

Co-clustering methods have been widely applied to document clustering and gene expression analysis. These methods make use of the duality between features and samples such that the co-occurring structure of sample and feature clusters can be extracted. In graph based co-clustering methods, a bipartite graph is constructed to depict the relation between features and samples. Most existing co-clustering methods conduct clustering on the graph achieved from the original data matrix, which doesn't have explicit cluster structure, thus they require a post-processing step to obtain the clustering results. In this paper, we propose a novel co-clustering method to learn a bipartite graph with exactly $k$ connected components, where $k$ is the number of clusters. The new bipartite graph learned in our model approximates the original graph but maintains an explicit cluster structure, from which we can immediately get the clustering results without post-processing. Extensive empirical results are presented to verify the effectiveness and robustness of our model.

## 1 Introduction

Clustering has long been a fundamental topic in unsupervised learning. The goal of clustering is to partition data into different groups. Clustering methods have been successfully applied to various areas, such as document clustering [3, 17], image segmentation [18, 7, 8] and bioinformatics [16, 14].

In clustering problems, the input data is usually formatted as a matrix, where one dimension represents samples and the other denotes features. Each sample can be seen as a data point characterized by a vector in the feature space. Alternatively, each feature can be regarded as a vector spanning in the sample space. Traditional clustering methods propose to cluster samples according to their distribution on features, or conversely, cluster features in terms of their distribution on samples.

In several types of data, such as document data and gene expression data, duality exists between samples and features. For example, in document data, we can reasonably assume that documents can be clustered based on their relations with different word clusters, while word clusters are formed according to their associations with distinct document clusters. However, in the one-sided clustering mechanism, the duality between samples and features is not taken into consideration. To make full use of the duality information, co-clustering methods (also known as bi-clustering methods) are proposed. The co-clustering mechanism takes advantage of the co-occurring cluster structure among features and samples to strengthen the clustering performance and gain better interpretation of the pragmatic meaning of the clusters.

Several co-clustering methods have been put forward to depict the relations between samples and features. In the graph based methods, the co-occurring structure between samples and features is usually treated as a bipartite graph, where the weights of edges indicate the relations between sample-feature pairs. In the left part of Fig. 1 we show an illustration of such bipartite graph, where the blue nodes on the left represent features while red nodes on the right show samples. The affinity between the features and samples is denoted by the weight of the corresponding edge. For example, $B_{ij}$ denotes the affinity between the $i$-th feature and the $j$-sample. In [4], the authors propose to minimize the cut between samples and features, which is equivalent to conducting spectral clustering on the bipartite graph. However, in this method, since the original graph doesn't display an explicit cluster structure, it still calls for the post-processing step like $K$-mean clustering to obtain the final clustering indicators, which may not be optimal.

To address this problem, in this paper, we propose a novel graph based co-clustering model to learn a bipartite graph with exactly $k$ connected components, where $k$ is the number of clusters. The new bipartite graph learned in our model approximates the original graph but maintains an explicit cluster structure, from which we can directly get the clustering results without post-processing steps. To achieve such an ideal structure of the new bipartite graph, we impose constraints on the rank of its Laplacian or normalized Laplacian matrix and derive algorithms to optimize the objective. We conduct several experiments to evaluate the effectiveness and robustness of our model. On both synthetic and benchmark datasets we gain equivalent or even better clustering results than other related methods.

**Notations:** Throughout the paper, all the matrices are written as uppercase. For matrix $M$, the $ij$-th element of $M$ is denoted by $m_{ij}$. The trace of matrix $M$ is denoted by $Tr(M)$. The $\ell_2$-norm of vector $v$ is denoted by $\|v\|_2$, the Frobenius norm of matrix $M$ is denoted by $\|M\|_F$.

## 2  Bipartite Spectral Graph Partitioning Revisited

The classic Bipartite Spectral Graph Partitioning (BSGP) method [4] is very effective for co-clustering. In order to simultaneously partition the rows and columns of a data matrix $B \in \mathbb{R}^{n_1 \times n_2}$, we first view $B$ as the weight matrix of a bipartite graph, where the left-side nodes are the $n_1$ rows of $B$, the right-side nodes are the $n_2$ columns of $B$, and the weight to connect the $i$-th left-side node and the $j$-th right-side node is $b_{ij}$ (see Fig.1). The procedure of BSGP is as follows:

**1)** Calculate $\tilde{A} = D_u^{-\frac{1}{2}} B D_v^{-\frac{1}{2}}$, where the diagonal matrices $D_u$ and $D_v$ are defined in Eq.(6).
**2)** Calculate $U$ and $V$, which are the leading $k$ left and right singular vectors of $\tilde{A}$, respectively.
**3)** Run the $K$-means on the rows of $F$ defined in Eq. (6) to obtain the final clustering results.

The bipartite graph can be viewed as an undirected weighted graph $\mathcal{G} = \{\mathcal{V}, A\}$ with $n = n_1 + n_2$ nodes, where $\mathcal{V}$ is the node set and the affinity matrix $A \in \mathbb{R}^{n \times n}$ is

$$A = \begin{bmatrix} 0 & B \\ B^T & 0 \end{bmatrix} \tag{1}$$

In the following, we will show that the BSGP method essentially performs spectral clustering with normalized cut on the graph $\mathcal{G}$.

Suppose the graph $\mathcal{G}$ is partitioned into $k$ components $\mathcal{V} = \{\mathcal{V}_1, \mathcal{V}_2, ..., \mathcal{V}_k\}$ . According to the spectral clustering, the normalized cut on the graph $\mathcal{G} = \{\mathcal{V}, A\}$ is defined as

$$\text{Ncut} = \sum_{i=1}^{k} \frac{cut(\mathcal{V}_i, \mathcal{V} \backslash \mathcal{V}_i)}{assoc(\mathcal{V}_i, \mathcal{V})} \tag{2}$$

where $\text{cut}(\mathcal{V}_i, \mathcal{V} \backslash \mathcal{V}_i) = \sum_{i \in \mathcal{V}_i, j \in \mathcal{V} \backslash \mathcal{V}_i} a_{ij}; \quad assoc(\mathcal{V}_i, \mathcal{V}) = \sum_{i \in \mathcal{V}_i, j \in \mathcal{V}} a_{ij}.$

Let $Y \in \mathbb{R}^{n \times k}$ be the partition indicator matrix, i.e., $y_{ij} = 1$ indicates the $i$-th node is partitioned into the $j$-th component. Then minimizing the normalized cut defined in Eq.(2) can be rewritten as the following problem:

$$\min_Y \sum_{i=1}^{k} \frac{y_i^T L y_i}{y_i^T D y_i} \tag{3}$$

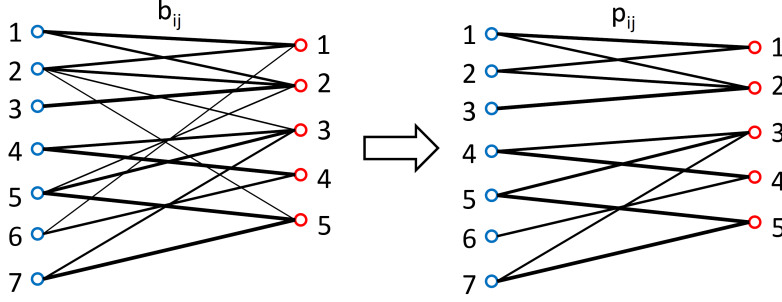

Figure 1: Illustration of the structured optimal bipartite graph.

where $y_i$ is the $i$-th column of $Y$, $L = D - A \in \mathbb{R}^{n \times n}$ is the Laplacian matrix, and $D \in \mathbb{R}^{n \times n}$ is the diagonal degree matrix defined as $d_{ii} = \sum_j a_{ij}$.

Let $Z = Y(Y^T D Y)^{-\frac{1}{2}}$, and denote the identity matrix by $I$, then problem (3) can be rewritten as

$$\min_{Z^T D Z = I} Tr(Z^T L Z) \tag{4}$$

Further, denotes $F = D^{\frac{1}{2}} Z = D^{\frac{1}{2}} Y(Y^T D Y)^{-\frac{1}{2}}$, then the problem (4) can be rewritten as

$$\min_{F^T F = I} Tr(F^T \tilde{L} F) \tag{5}$$

where $\tilde{L} = I - D^{-\frac{1}{2}} A D^{-\frac{1}{2}}$ is the normalized Laplacian matrix.

We rewrite $F$ and $D$ as the following block matrices:

$$F = \begin{bmatrix} U \\ V \end{bmatrix}, D = \begin{bmatrix} D_u & \\ & D_v \end{bmatrix} \tag{6}$$

where $U \in \mathbb{R}^{n_1 \times k}$, $V \in \mathbb{R}^{n_2 \times k}$, $D_u \in \mathbb{R}^{n_1 \times n_1}$, $D_v \in \mathbb{R}^{n_2 \times n_2}$.

Then according to the definition of $A$ in Eq. (1), the problem (5) can be further rewritten as

$$\max_{U^T U + V^T V = I} Tr(U^T D_u^{-\frac{1}{2}} B D_v^{-\frac{1}{2}} V) \tag{7}$$

Note that in addition to the constraint $U^T U + V^T V = I$, the $U, V$ should be constrained to be discrete values according to the definitions of $U$ and $V$. This discrete constraint makes the problem very difficult to solve. To address it, we first remove the discrete constraint to make the problem (7) solvable with Lemma 1, and then run $K$-means on $U$ and $V$ to get the discrete solution.

**Lemma 1** *Suppose $M \in \mathbb{R}^{n_1 \times n_2}$, $X \in \mathbb{R}^{n_1 \times k}$, $Y \in \mathbb{R}^{n_2 \times k}$. The optimal solutions to the problem*

$$\max_{X^T X + Y^T Y = I} Tr(X^T M Y) \tag{8}$$

*are $X = \frac{\sqrt{2}}{2} U_1$, $Y = \frac{\sqrt{2}}{2} V_1$, where $U_1, V_1$ are the leading $k$ left and right singular vectors of $M$, respectively.*

**Proof:** Denote the Lagrangian function of the problem is $\mathcal{L}(X, Y, \Lambda) = Tr(X^T A Y) - Tr(\Lambda(X^T X + Y^T Y - I))$ By setting the derivative of $\mathcal{L}(X, Y, \Lambda)$ w.r.t. $X$ to zero, we have $AY = X\Lambda$. By taking the derivative of $\mathcal{L}(X, Y, \Lambda)$ w.r.t. $Y$ to zero, we have $A^T X = Y\Lambda$. Thus $AA^T X = AY\Lambda = X\Lambda^2$. Therefore, the optimal solution $X$ should be the eigenvectors of $AA^T$, i.e, the left singular vectors of $M$. Similarly, the optimal solution $Y$ should be the right singular vectors of $M$. Since it is a maximization problem, the optimal solution $X, Y$ should be the leading $k$ left and right singular vectors of $M$, respectively. $\square$

According to Lemma 1, if the discrete constraint on $U$ and $V$ is not considered, the optimal solution $U$ and $V$ to the problem (7) are the leading $k$ left and right singular vectors of $\tilde{A} = D_u^{-\frac{1}{2}} B D_v^{-\frac{1}{2}}$, respectively.

Since the solution $U$ and $V$ are not discrete values, we need to run the $K$-means on the rows of $F$ defined in Eq.(6) to obtain the final clustering results.

# 3 Learning Structured Optimal Bipartite Graph for Co-Clustering

## 3.1 Motivation

We can see from the previous section that the given $B$ or $A$ does not have a very clear clustering structure (i.e., $A$ is not a block diagonal matrix with proper permutation) and the $U$ and $V$ are not discrete values, thus we need run the $K$-means to obtain the final clustering results. However, $K$-means is very sensitive to the initialization, which makes the clustering performance unstable and suboptimal.

To address this challenging and fundamental problem, we target to learn a new graph similarity matrix $S \in \mathbb{R}^{n \times n}$ or $P \in \mathbb{R}^{n_1 \times n_2}$ as

$$S = \left[ \begin{array}{cc} 0 & P \\ P^T & 0 \end{array} \right], \tag{9}$$

such that the new graph is more suitable for clustering task. In our strategy, we learn an $S$ that has exact $k$ connected components, see Fig. 1. Obviously such a new graph can be considered as the ideal graph for clustering task with providing clear clustering structure. If $S$ has exact $k$ connected components, we can directly obtain the final clustering result based on $S$, without running $K$-means or other discretization procedures as traditional graph based clustering methods have to do.

The learned structured optimal graph similarity matrix $S$ should be as close as possible to the given graph affinity matrix $A$, so we propose to solve the following problem:

$$\min_{P \geq 0, P1=1, S \in \Omega} \|S - A\|_F^2 \tag{10}$$

where $\Omega$ is the set of matrices $S \in \mathbb{R}^{n \times n}$ which have exact $k$ connected components.

According to the special structure of $A$ and $S$ in Eq. (1) and Eq. (9), the problem (10) can be written as

$$\min_{P \geq 0, P1=1, S \in \Omega} \|P - B\|_F^2 \tag{11}$$

The problem (11) seems very difficult to solve since the constraint $S \in \Omega$ is intractable to handle. In the next subsection, we will propose a novel and efficient algorithm to solve this problem.

## 3.2 Optimization

If the similarity matrix $S$ is nonnegative, then the Laplacian matrix $L_S = D_S - S$ associated with $S$ has an important property as follows [13, 12, 11, 2].

**Theorem 1** *The multiplicity $k$ of the eigenvalue 0 of the Laplacian matrix $L_S$ is equal to the number of connected components in the graph associated with $S$.*

Theorem 1 indicates that if $rank(L_S) = n - k$, the constraint $S \in \Omega$ will be held. Therefore, the problem (11) can be rewritten as:

$$\min_{P \geq 0, P1=1, rank(L_S)=n-k} \|P - B\|_F^2 \tag{12}$$

Suppose $\sigma_i(L_S)$ is the $i$-th smallest eigenvalue of $L_S$. Note that $\sigma_i(L_S) \geq 0$ because $L_S$ is positive semi-definite. The problem (12) is equivalent to the following problem for a large enough $\lambda$:

$$\min_{P \geq 0, P1=1} \|P - B\|_F^2 + \lambda \sum_{i=1}^{k} \sigma_i(L_S) \tag{13}$$

When $\lambda$ is large enough (note that $\sigma_i(L_S) \geq 0$ for every $i$), the optimal solution $S$ to the problem (13) will make the second term $\sum_{i=1}^{k} \sigma_i(L_S)$ to be zero, and thus the constraint $rank(L_S) = n - k$ in the problem (12) would be satisfied.

According to the Ky Fan's Theorem [6], we have:

$$\sum_{i=1}^{k} \sigma_i(L_S) = \min_{F \in \mathbb{R}^{n \times k}, F^T F = I} Tr(F^T L_S F) \tag{14}$$

Therefore, the problem (13) is further equivalent to the following problem

$$\min_{P,F} \|P - B\|_F^2 + \lambda Tr(F^T L_S F)$$
$$s.t. \quad P \geq 0, P1 = 1, F \in \mathbb{R}^{n \times k}, F^T F = I \tag{15}$$

The problem (15) is much easier to solve compared with the rank constrained problem (12). We can apply the alternating optimization technique to solve this problem.

When $P$ is fixed, the problem (15) becomes:

$$\min_{F \in \mathbb{R}^{n \times k}, F^T F = I} Tr(F^T L_S F) \tag{16}$$

The optimal solution $F$ is formed by the $k$ eigenvectors of $L_S$ corresponding to the $k$ smallest eigenvalues.

When $F$ is fixed, the problem (15) becomes

$$\min_{P \geq 0, P1 = 1} \|P - B\|_F^2 + \lambda Tr(F^T L_S F) \tag{17}$$

According to the property of Laplacian matrix, we have the following relationship:

$$Tr(F^T L_S F) = \frac{1}{2} \sum_{i=1}^{n} \sum_{j=1}^{n} \|f_i - f_j\|_2^2 \, s_{ij} \tag{18}$$

where $f_i$ is the $i$-th row of $F$.

Thus according to the structure of $S$ defined in Eq.(9), Eq.(18) can be rewritten as

$$Tr(F^T L_S F) = \sum_{i=1}^{n_1} \sum_{j=1}^{n_2} \|f_i - f_j\|_2^2 \, p_{ij} \tag{19}$$

Based on Eq. (19), the problem (17) can be rewritten as

$$\min_{P \geq 0, P1 = 1} \sum_{i=1}^{n_1} \sum_{j=1}^{n_2} (p_{ij} - b_{ij})^2 + \lambda \|f_i - f_j\|_2^2 \, p_{ij} \tag{20}$$

Note that the problem (20) is independent between different $i$, so we can solve the following problem individually for each $i$. Denote $v_{ij} = \|f_i - f_j\|_2^2$, and denote $v_i$ as a vector with the $j$-th element as $v_{ij}$ (same for $p_i$ and $b_i$), then for each $i$, the problem (20) can be written in the vector form as

$$\min_{p_i^T \mathbf{1} = 1, p_i \geq \mathbf{0}} \left\| p_i - \left( b_i - \frac{\lambda}{2} v_i \right) \right\|_2^2 \tag{21}$$

This problem can be solved by an efficient iterative algorithm [9].

The detailed algorithm to solve the problem (15) is summarized in Algorithm 1. In the algorithm, we can only update the $m$ nearest similarities for each data points in $P$ and thus the complexity of updating $P$ and updating $F$ (only need to compute top $k$ eigenvectors on very sparse matrix) can be reduced significantly. Nevertheless, Algorithm 1 needs to conduct eigen-decomposition on an $n \times n(n = n_1 + n_2)$ matrix in each iteration, which is time consuming. In the next section, we will propose another optimization algorithm, which only needs to conduct SVD on an $n_1 \times n_2$ matrix in each iteration, and thus is much more efficient than Algorithm 1.

---

**Algorithm 1** Algorithm to solve the problem (15).

---

**input** $B \in \mathbb{R}^{n_1 \times n_2}$, cluster number $k$, a large enough $\lambda$.
**output** $P \in \mathbb{R}^{n_1 \times n_2}$ and thus $S \in \mathbb{R}^{n \times n}$ defined in Eq.(9) with exact $k$ connected components.
Initialize $F \in \mathbb{R}^{n \times k}$, which is formed by the $k$ eigenvectors of $L = D - A$ corresponding to the $k$ smallest eigenvalues, $A$ is defined in Eq. (1).
**while** not converge **do**
    1. For each $i$, update the $i$-th row of $P$ by solving the problem (21), where the $j$-th element of $v_i$ is $v_{ij} = \|f_i - f_j\|_2^2$.
    2. Update $F$, which is formed by the $k$ eigenvectors of $L_S = D_S - S$ corresponding to the $k$ smallest eigenvalues.
**end while**

---

# 4 Speed Up the Model

If the similarity matrix $S$ is nonnegative, then the normalized Laplacian matrix $\tilde{L}_S = I - D_S^{-\frac{1}{2}} S D_S^{-\frac{1}{2}}$ associated with $S$ also has an important property as follows [11, 2].

**Theorem 2** *The multiplicity $k$ of the eigenvalue 0 of the normalized Laplacian matrix $\tilde{L}_S$ is equal to the number of connected components in the graph associated with $S$.*

Theorem 2 indicates that if $rank(\tilde{L}_S) = n - k$, the constraint $S \in \Omega$ will be hold. Therefore, the problem (11) can also be rewritten as

$$\min_{P \geq 0, P1=1, rank(\tilde{L}_S)=n-k} \|P - B\|_F^2 \tag{22}$$

Similarly, the problem (22) is equivalent to the following problem for a large enough value of $\lambda$:

$$\min_{P,F} \|P - B\|_F^2 + \lambda Tr(F^T \tilde{L}_S F) \\ s.t. \quad P \geq 0, P1 = 1, F \in \mathbb{R}^{n \times k}, F^T F = I \tag{23}$$

Again, we can apply the alternating optimization technique to solve problem (23).

When $P$ is fixed, since $\tilde{L}_S = I - D_S^{-\frac{1}{2}} S D_S^{-\frac{1}{2}}$, the problem (23) becomes

$$\max_{F \in \mathbb{R}^{n \times k}, F^T F = I} Tr(F^T D_S^{-\frac{1}{2}} S D_S^{-\frac{1}{2}} F) \tag{24}$$

We rewrite $F$ and $D_S$ as the following block matrices:

$$F = \begin{bmatrix} U \\ V \end{bmatrix}, \qquad D_S = \begin{bmatrix} D_{Su} & \\ & D_{Sv} \end{bmatrix} \tag{25}$$

where $U \in \mathbb{R}^{n_1 \times k}, V \in \mathbb{R}^{n_2 \times k}, D_{Su} \in \mathbb{R}^{n_1 \times n_1}, D_{Sv} \in \mathbb{R}^{n_2 \times n_2}$.

Then according to the definition of $S$ in Eq. (9), the problem (24) can be further rewritten as

$$\max_{U^T U + V^T V = I} Tr(U^T D_{Su}^{-\frac{1}{2}} P D_{Sv}^{-\frac{1}{2}} V) \tag{26}$$

According to Lemma 1, the optimal solution $U$ and $V$ to the problem (26) are the leading $k$ left and right singular vectors of $\tilde{S} = D_{Su}^{-\frac{1}{2}} P D_{Sv}^{-\frac{1}{2}}$, respectively.

When $F$ is fixed, the problem (23) becomes

$$\min_P \|P - B\|_F^2 + \lambda Tr(F^T \tilde{L}_S F) \\ s.t. \quad P \geq 0, P1 = 1 \tag{27}$$

According to the property of normalized Laplacian matrix, we have the following relationship:

$$Tr(F^T \tilde{L}_S F) = \frac{1}{2} \sum_{i=1}^{n} \sum_{j=1}^{n} \left\| \frac{f_i}{\sqrt{d_i}} - \frac{f_j}{\sqrt{d_j}} \right\|_2^2 s_{ij} \tag{28}$$

Thus according to the structure of $S$ defined in Eq.(9), and denote $v_{ij} = \left\| \frac{f_i}{\sqrt{d_i}} - \frac{f_j}{\sqrt{d_j}} \right\|_2^2$, the problem (27) can be rewritten as

$$\min_{P \geq 0, P1=1} \sum_{i=1}^{n_1} \sum_{j=1}^{n_2} (p_{ij} - b_{ij})^2 + \lambda v_{ij} p_{ij},$$

which has the same form as in Eq. (20) and thus can be solved efficiently.

The detailed algorithm to solve the problem (23) is summarized in Algorithm 2. In the algorithm, we can also only update the $m$ nearest similarities for each data points in $P$ and thus the complexity of updating $P$ and updating $F$ can be reduced significantly.

Note that Algorithm 2 only needs to conduct SVD on an $n_1 \times n_2$ matrix in each iteration. In some cases, $\min(n_1, n_2) \ll (n_1 + n_2)$, thus Algorithm 2 is much more efficient than Algorithm 1. Therefore, in the next section, we use Algorithm 2 to conduct the experiments.

---

**Algorithm 2** Algorithm to solve the problem (23).

---

**input** $B \in \mathbb{R}^{n_1 \times n_2}$, cluster number $k$, a large enough $\lambda$.
**output** $P \in \mathbb{R}^{n_1 \times n_2}$ and thus $S \in \mathbb{R}^{n \times n}$ defined in Eq.(9) with exact $k$ connected components.
Initialize $F \in \mathbb{R}^{n \times k}$, which is formed by the $k$ eigenvectors of $\tilde{L} = I - D^{-\frac{1}{2}} A D^{-\frac{1}{2}}$ corresponding to the $k$ smallest eigenvalues, $A$ is defined in Eq. (1).
**while** not converge **do**
    1. For each $i$, update the $i$-th row of $P$ by solving the problem (21), where the $j$-th element of $v_i$ is $v_{ij} = \left\| \frac{f_i}{\sqrt{d_i}} - \frac{f_j}{\sqrt{d_j}} \right\|_2^2$.
    2. Update $F = \begin{bmatrix} U \\ V \end{bmatrix}$, where $U$ and $V$ are the leading $k$ left and right singular vectors of $\tilde{S} = D_{Su}^{-\frac{1}{2}} P D_{Sv}^{-\frac{1}{2}}$ respectively and $D_S = \begin{bmatrix} D_{Su} & \\ & D_{Sv} \end{bmatrix}$.
**end while**

---

# 5 Experimental Results

In this section, we conduct multiple experiments to evaluate our model. We will first introduce the experimental settings throughout the section and then present evaluation results on both synthetic and benchmark datasets.

## 5.1 Experimental Settings

We compared our method (denoted by SOBG) with two related co-clustering methods, including Bipartite Spectral Graph Partition (BSGP) [4] and Orthogonal Nonnegative Matrix Tri-Factorizations (ONMTF) [5]. Also, we introduced several one-sided clustering methods to the comparison, which are $K$-means clustering, Normalized Cut (NCut) and Nonnegative Matrix Factorization (NMF).

For methods requiring a similarity graph as the input, *i.e.,* NCut and NMF, we adopted the self-tuning Gaussian method [19] to construct the graph, where the number of neighbors was set to be 5 and the $\sigma$ value was self-tuned. In the experiment, there are four methods involving $K$-means clustering, which are $K$-means, NCut, BSGP and ONMTF (the latter three methods need $K$-means as the post-processing step to get the clustering results). When running $K$-means we used 100 random initializations for all these four methods and recorded the average performance over these 100 runs as well as the best one with respect to the $K$-means objective function value.

In our method, to accelerate the algorithmic procedure, we determined the parameter $\lambda$ in an heuristic way: first specify the value of $\lambda$ with an initial guess; next, we computed the number of zero eigenvalues in $\tilde{L}_S$ in each iteration, if it was larger than $k$, then divided $\lambda$ by 2; if smaller then multiplied $\lambda$ by 2; otherwise we stopped the iteration.

The number of clusters was set to be the ground truth. The evaluation of different methods was based on the percentage of correctly clustered samples, *i.e.,* clustering accuracy.

## 5.2 Results on Synthetic Data

In this subsection, we first apply our method to the synthetic data as a sanity check. The synthetic data is constructed as a two-dimensional matrix, where rows and columns come from three clusters respectively. Row clusters and column clusters maintain mutual dependence, *i.e.,* rows and columns from the first cluster form a block along the diagonal of the data matrix, and this also holds true for the second and third cluster. The number of rows for each cluster is 20, 30 and 40 respectively, while the number of columns is 30, 40 and 50. Each block is generated randomly with elements i.i.d. sampled from Gaussian distribution $\mathcal{N}(0, 1)$. Also, we add noise to the "non-block" area of the data matrix, *i.e.,* all entries in the matrix excluding elements in the three clusters. The noise can be denoted as $r * \delta$, where $\delta$ is Gaussian noise i.i.d. sampled from Gaussian distribution $\mathcal{N}(0, 1)$ and $r$

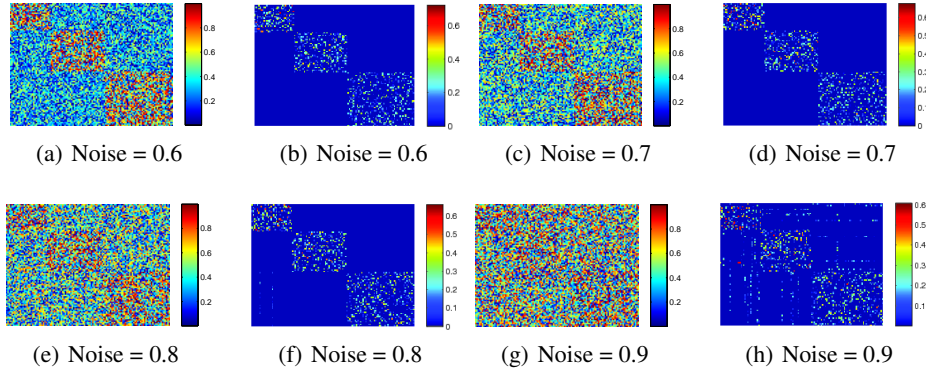

| | (a) Noise = 0.6 | (b) Noise = 0.6 | (c) Noise = 0.7 | (d) Noise = 0.7 |
| (e) Noise = 0.8 | (f) Noise = 0.8 | (g) Noise = 0.9 | (h) Noise = 0.9 |

Figure 2: Illustration of the data matrix in different settings of noise. Different rows of figures come from different settings of noise. In each row, figures on the left column are the original data matrices generated in the experiment, while on the right column display the bipartite matrix $B$ learned in our model which approximates the original data matrix and maintains the block structure.

| | Methods | Noise = 0.6 | Noise = 0.7 | Noise = 0.8 | Noise = 0.9 |
|---|---|---|---|---|---|
| | $K$-means | 99.17 | 97.50 | 71.67 | 39.17 |
| | NCut | 99.17 | 95.00 | 46.67 | 38.33 |
| Clustering | NMF | 98.33 | 95.00 | 46.67 | 37.50 |
| Accuracy(%) | BSGP | **100.00** | 93.33 | 62.50 | 40.00 |
| on Rows | ONMTF | 99.17 | 97.50 | 71.67 | 39.17 |
| | SOBG | **100.00** | **100.00** | **98.33** | **84.17** |
| | $K$-means | **100.00** | 95.56 | 51.11 | 46.67 |
| | NCut | **100.00** | 91.11 | 60.00 | 38.89 |
| Clustering | NMF | **100.00** | 90.00 | 47.78 | 37.78 |
| Accuracy(%) | BSGP | **100.00** | 93.33 | 63.33 | 46.67 |
| on Columns | ONMTF | **100.00** | 95.56 | 51.11 | 46.67 |
| | SOBG | **100.00** | **100.00** | **100.00** | **87.78** |

Table 1: Clustering accuracy comparison on rows and columns of the synthetic data in different portion of noise.

is the portion of noise. We set $r$ to be {0.6, 0.7, 0.8, 0.9} respectively so as to evaluate the robustness of different methods under the circumstances of various disturbance.

We apply all comparing methods to the synthetic data and assess their ability to cluster the rows and columns. One-sided clustering methods are applied to the data twice (once to cluster rows and the other time to cluster columns) such that clustering accuracy on these two dimensions can be achieved. Co-clustering methods can obtain clustering results on both dimensions simultaneously in one run.

In Table 1 we summarize the clustering accuracy comparison on both rows and columns under different settings of noise. In Fig. 2 we display the corresponding original data matrix and the bipartite matrix $B$ learned in our model. We can notice that when the portion of noise $r$ is relatively low, *i.e.,* 0.6 and 0.7, the block structure of the original data is clear, then all methods perform fairly well in clustering both rows and columns. However, as $r$ increases, the block structure in the original data blurs thus brings obstacles to the clustering task. With high portion of noise, all other methods seem to be disturbed to a large extent while our method shows apparent robustness. Even when the portion of noise becomes as high as 0.9, such that the structure of clusters in the original data becomes hard to distinguish with eyes, our method still excavates a reasonable block arrangement with a clustering accuracy of over 80%. Also, we can find that co-clustering methods usually outperform one-sided clustering methods since they utilize the interrelations between rows and columns. The interpretation of the co-clustering structure strengthens the performance, which conforms to our theoretical analysis.

| Methods | | Reuters21578 | LUNG | Prostate-MS | prostateCancerPSA410 |
|---|---|---|---|---|---|
| $K$-means | Ave | 40.86±4.59 | 61.91±6.00 | 46.47±3.26 | 64.15±9.40 |
| | Best | 32.77 | 71.43 | 45.34 | 62.92 |
| NCut | Ave | 26.92±0.93 | 69.67±14.26 | 46.86±1.19 | 55.06±0.00 |
| | Best | 29.18 | **79.80** | 47.20 | 55.06 |
| NMF | | 30.91 | 75.86 | 47.83 | 55.06 |
| BSGP | Ave | 11.44±0.39 | 64.95±5.06 | 46.27±0.00 | 57.30±0.00 |
| | Best | 11.26 | 70.94 | 46.27 | 57.30 |
| ONMTF | Ave | 17.57±1.95 | 61.31±10.34 | 45.46±3.18 | 62.92±0.00 |
| | Best | 27.90 | 71.43 | 45.34 | 62.92 |
| SOBG | | **43.94** | **78.82** | **62.73** | **69.66** |

Table 2: Clustering accuracy comparison on four benchmark datasets. For the four methods involving $K$-means clustering, *i.e.,* $K$-means, NCut, BSGP and ONMTF, their average performance (Ave) over 100 repetitions and the best one (Best) w.r.t. $K$-means objective function value were both reported.

## 5.3   Results on Benchmark Data

In this subsection, we use four benchmark datasets for the evaluation. There are one document dataset and three gene expression datasets participating in the experiment, the property of which is introduced in details as below.

**Reuters21578** dataset is processed and downloaded from `http://www.cad.zju.edu.cn/home/dengcai/Data/TextData.html`. It contains 8293 documents in 65 topics. Each document is depicted by its frequency on 18933 terms.

**LUNG** dataset [1] provides a source for the study of lung cancer. It has 203 samples in five classes, among which there are 139 adenocarcinoma (AD), 17 normal lung (NL), 6 small cell lung cancer (SMCL), 21 squamous cell carcinoma (SQ) as well as 20 pulmonary carcinoid (COID) samples. Each sample has 3312 genes.

**Prostate-MS** dataset [15] contains a total of 332 samples from three different classes, which are 69 samples diagnosed as prostate cancer, 190 samples of benign prostate hyperplasia, as well as 63 normal samples showing no evidence of disease. Each sample has 15154 genes.

**ProstateCancerPSA410** dataset [10] describes gene information of patients with prostate-specific antigen (PSA)-recurrent prostate cancer. It includes a total of 89 samples from two classes. Each sample has 15154 genes.

Before the clustering process, feature scaling was performed on each dataset such that features are on the same scale of [0, 1]. Also, the $\ell_2$-norm of each feature was normalized to 1.

Table 2 summarizes the clustering accuracy comparison on these benchmark datasets. Our method performs equally or even better than the alternatives on all these datasets. This verifies the effectiveness of our method in the practical situation. There is an interesting phenomenon that the advantage of our method tends to be more obvious for higher dimensional data. This is because high-dimensional features make the differences in the distance between samples to be smaller thus the cluster structure of the original data becomes vague. In this case, since our model is more robust compared with the alternative methods (verified in the synthetic experiments), we can get better clustering results.

## 6   Conclusions

In this paper, we proposed a novel graph based co-clustering model. Different from existing methods which conduct clustering on the graph achieved from the original data, our model learned a new bipartite graph with explicit cluster structure. By imposing the rank constraint on the Laplacian matrix of the new bipartite graph, we guaranteed the learned graph to have exactly $k$ connected components, where $k$ is the number of clusters. From this ideal structure of the new bipartite graph learned in our model, the obvious clustering structure can be obtained without resorting to post-processing steps. We presented experimental results on both synthetic data and four benchmark datasets, which validated the effectiveness and robustness of our model.

## Footnotes

*This work was partially supported by U.S. NSF-IIS 1302675, NSF-IIS 1344152, NSF-DBI 1356628, NSF-IIS 1619308, NSF-IIS 1633753, NIH AG049371.

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
