[Reviews · NeurIPS 2017]

Reviewer 1



This paper achieves co-clustering by formulating the problem as optimization of a graph similarity matrix with k components. This paper is well motivated, the formulation of the proposed method is interesting, and its empirical performance is superior to other co-clustering methods. However, I have the following concerns about the clarity and the technical soundness of the paper. - Do Algorithm 1 and 2 always converge to a global solution? There is no theoretical guarantee of it, which is important to show the effectiveness of the proposed algorithms. - Section 4 is almost the same as Section 3.2, and the only difference is that the Laplacian matrix is normalized. This is confusing and the readability is not high. I recommend to integrate Section 4 into Section 3.2. - Moreover, the title of Section 4 is "Speed Up the Model", but how to speed up is not directly explained. - Algorithms 1 and 2 assume that lambda is large enough. But this assumption is not used in experiments. In addition, in experiments lambda is divided or multiplied by 2 until convergence, which is not explained in Algorithms 1 and 2, hence the explanation is not consistent and confusing. - Clustering accuracy is reported as 100% when noise = 0.6. However, in Figure 2, there are blues pixels in clusters and illustrated results do not look like 100% accuracy. More detailed explanation is helpful. - In Table 2, there is a large difference of the accuracy between the proposed method and BSGP, although the strategy is similar. What makes the difference? Discussing the reason would be interesting.

Reviewer 2



The authors propose a new method for co-clustering. The idea is to learn a bipartite graph with exactly k connected components. This way, the clusters can be directly inferred and no further preprocessing step (like executing k-means) is necessary. After introducing their approach the authors conduct experiments on a synthetic data set as well as on four benchmark data sets. I think that the proposed approach is interesting. However, there are some issues. First, it is not clear to me how the synthetic data was generated. Based on the first experiment I suppose it is too trivial anyway. Why did the authors not use k-means++ instead of standard k-means (which appears in 3/5 baselines)? Why did they not compare against standard spectral clustering? How did they initialize NMF? Randomly? Why not 100 random initializations like in the other methods? Why did they not use a second or third evaluation measure like the adjusted rand index or normalized mutual information? Since the authors introduce another parameter lambda (beside k) I would like to see the clustering performance in dependence of lambda to see how critical the choice of lambda really is. I see the argument why Algorithm 2 is faster than Algorithm 1, but I would like to see how they compare within the evaluation, because I doubt that they perform exactly the same. How much faster is Algorithm 2 compared to Algorithm 1? Second, the organization of the paper has to be improved. Figure 1 is referenced already on the first page but appears on the third page. The introduced baseline BSGP should be set as an algorithm on page two. Furthermore, the authors tend to use variables before introducing them, i.e., D_u, D_v and F in lines 61-63. Regarding the title: In what sense is the approach optimal? Isn't bipartite and structured the same in your case? Third, the language of the paper should be improved. There are some weird sentence constructions, especially when embedded sentences are present. Furthermore, there are too many "the", singular/plural mistakes and a problem with tenses, i.e. Section 5.2 should be in present tense. In addition to grammar, the mathematical presentation should be improved. Equations are part of the sentences and hence, need punctation marks. Due to the inadequate evaluation and the quality of the write-up, I vote for rejecting the paper. More detailed comments: *** Organization: - title: I do not get where the "optimal" comes from. Lines 33-34 say Bipartite=Structured, so why are both words within the title? - line 35: Figure 1 should be on the top of page two - lines 61-63 shoud appear as an algorithm - line 61: in order to understand the matrices D_u and D_v, the reader has to jump to line 79 and then search backwards for D, which is defined in line 74.. - line 63: define D_u, D_v and F before using them.. *** Technical: - line 35: B is used for a graph but also for the data matrix (lines 35 and 57) - line 38: B_{ij} should be small (see lines 53 and 60) - almost all equations: math is part of the text and therefore punctuation marks (.,) have to be used - equation (2) and line 70, set Ncut, cut and assoc with \operatorname{} - line 74: order: first introduce D, then d_{ii} and then L - line 74: if D is a diagonal matrix, then D_u and D_v are diagonal too. Say it! - equation (4): introduce I as the identity matrix - lines 82-85: this paragraph is problematic. What is a discrete value? Should the entries of U and V be integer? ( I think there a just a few orthogonal matrices which are integer ) How does someone obtain a discrete solution of by running k-means on U and V? - line 93: if P (and therefore S) should be non-negative (see Equation (9)), then you can already say it in line 93. Furthermore, isn't this an assumption you have to make? Until line 93, A and B could contain negative numbers but in the proposed approach, this is no longer possible, right? - line 113: a sentence does not start with a variable, in this case \sigma - line 132: get rid of (n=n_1+n_2) or add a where before - line 149: \min - line 150: Algorithms 1 and 2 are working on different graph laplacians. I understand that Alg.2 is faster than Alg.1 but shouldn't there be also a difference in performance? What are the results? It should be noted if Alg.1 performs exactly the same as Alg.2. However, another graph laplacian should make a difference. - line 160: Why does NMF need a similarity graph? - line 161: Did you tune the number of neighbors? - line 164: Why did you not use k-means++? - line 167: How does the performance of the proposed method vary with different values for the newly introduced parameter lambda? - line 171: How does the performance of the proposed method vary with different values for k. Only the ground-truth k was evaluated.. - line 175: "two-dimensional matrix" Why not $n_1 \times n_2$? - Section 5.2: Please improve the description on how the synthetic data was generated. - line 182: if \delta is iid N(0,1) and you scale it with r greater than 0, then r*\delta is N(0,r^2) - line 183: according to the values of r, the noise was always less than the "data"? - Figure 2: Improve layout and caption. - Table 1: After having 100 executions of all algorithms, what are the standard deviations or standard errors of the values within the table? - line 183: r is not the portion of noise - line 183: you did not do a evaluation of robustness! - line 194: setting r=0.9 does not yield a high portion of noise. Perhaps the description of the synthetic data needs improvement. - Table 2: NMF has to be randomly initialized as well, hence I am missing the average over 100 executions - Table 2: What is +-4.59? One standard deviation? Standard error? Something else? - line 219: You forgot to mention the obvious limitation of your approach: you assume the data to be non-negative! - line 220: \ell_2-norm vs. L2-norm (line 53) - line 224: Please elaborate the sentence "This is because high-dimensional..." *** Language: - line 4: graph-based - line 20: feature space - line 22: distribution (singular) - line 27: the duality [...] is - line 33: In graph-based - line 35: of such a bipartite graph - line 37: The affinity [...] is - line 38: propose, stay in present tense - line 39: conduct - line 40: "in this method" Which method is meant? - line 43: graph-based - lines 53-54: L2-norm vs. Frobenius norm, with or without - - line 58: "view" - line 66: In the following, - line 72: component (singular) - lines 76-77: "denotes" is wrong, better use "Let Z=" - line 82: Note that - line 88: We can see from the previous section - lines 88-90: this is a strange sentence - line 93: we learn a matrix S that has - line 96: If S has exactly k connected - line 101: S, which have exactly k connected - lines 104-105: In the next subsection - line 132: to conduct an eigen-decomposition - line 134: to conduct an SVD - Algorithm 1: while not converged - line 141: "will be hold" - line 148: to conduct an SVD - line 149: "Algorithm 2 is much efficient than Algorithm 1" - Section 5.1: the section should be written in present tense (like Section 5.2) not in past tense - line 159: Non-negative Matrix Factorization (NMF) - line 204: data sets do not participate in experiments - line 229: graph-based - line 232: "we guaranteed the learned grapg to have explicitly k connected components" - unnecessary use of "the" in lines 33, 68, 76, 77, 81, 82, 102, 104, 111, 114, 118, 119, 120, 122, 125, 129, 141, 143, 145